

# Preventive behaviors of COVID-19 during the COVID-19 pandemic among community-dwelling older adults in Thailand

Kanchana Piboon[1], Jarinthip Chomchaipon[2], Dhammawat Ouppawongsapat[3], Wanlop Jaidee[3], Patchana Hengboriboonpong Jaidee[3], Paiboon Pongsaengpan[3] and Wiriya Mahikul[4]

[1] Faculty of Nursing, Burapha University, Chon Buri, Thailand
[2] Faculty of Science and Technology, Sakon Nakhon Rajabhat University, Sakon Nakhon, Thailand
[3] Faculty of Public Health, Burapha University, Chon Buri, Thailand
[4] Princess Srisavangavadhana College of Medicine, Chulabhorn Royal Academy, Bangkok, Thailand

Corresponding author
Wiriya Mahikul,
wiriya.mah@cra.ac.th

## ABSTRACT

**Background**. The COVID-19 pandemic was a major public health crisis, especially among older people. This study aimed to examine factors affecting preventive behaviors among community-dwelling older adults across all regions of Thailand during the COVID-19 pandemic using the health belief model (HBM).

**Methods**. This cross-sectional study included 910 participants from Thailand in July and August 2021. A multistage stratified random sampling technique was used to select participants. Data were collected through a structured interview process. Data analysis was conducted using multiple linear regression.

**Results**. The participants' mean age was $66.5 \pm 4.64$ years, with the majority being female (61.8%) and residing in the central region (26.5%). The results indicated that all participants (100.0%) agreed that COVID-19 can be prevented by personal protective equipment, such as masks and disposable gloves. Furthermore, the participants' adherence to COVID-19 preventive measures was evaluated, revealing that the majority (55.8%) always practiced hand hygiene by washing hands with alcohol gel or soap and cleaning them with water before eating. Regression analysis indicated that COVID-19 preventive behaviors were significantly associated with knowledge ($b = 0.091$), perceived susceptibility ($b = 0.066$), perceived benefits ($b = 0.111$), perceived barriers ($b = -0.040$), and cues to action ($b = 0.110$) with $p < 0.01$.

**Conclusions**. Increased knowledge, perceived susceptibility, perceived benefits, cues to action, and decreased perceived barriers scores were associated with higher practice scores among community-dwelling older adults during the COVID-19 pandemic in Thailand. To improve practices, health information campaigns should focus on highlighting the advantages of preventive behaviors, offering tips and advice to overcome barriers, providing cues to action through various reminders on social media, and increasing awareness about disease prevention and control in future pandemics or new disease outbreaks.

# INTRODUCTION

The COVID-19 pandemic was a global health crisis (*Arshad Ali et al., 2020*; *Mallah et al., 2021*; *Mofijur et al., 2021*). Older adults, in particular, had a high risk of severe illness and dying from the virus (*Lebrasseur et al., 2021*; *Zhu, Liu & Jiang, 2022*). As of April 6, 2023, the total number of COVID-19 cases reported globally was 762,128,709, with 6,892,741 cumulative deaths (*WHO, 2023*). Thailand has reported 4,728,967 confirmed cases of COVID-19 and 33,940 deaths (*WHO Thailand, 2023*). As of May 14, 2022, a total of 527,878 cases and 21,043 deaths (71.4%) were recorded among elderly in Thailand, making this age group the most affected by mortality (*Napalai et al., 2022*). Furthermore, most elderly people infected with COVID-19 had underlying chronic non-communicable diseases (NCDs), such as diabetes, hypertension, obesity, chronic obstructive pulmonary disease (COPD), ischemic heart disease, stroke, and cancer (*Napalai et al., 2022*).

Various measures have been implemented to reduce COVID-19 transmission and death rates, such as promptly identifying suspected cases; rapidly testing and isolating new cases; tracing contacts; implementing quarantines, travel, and mobility restrictions; and promoting individual-level nonpharmaceutical interventions, such as physical distancing, hand washing, wearing masks, working from home, and encouraging active involvement of communities (*Karimy et al., 2021*; *Sohrabi et al., 2020*). In Thailand, the government implemented measures such as social distancing, mandatory preventive behaviors, travel restrictions, case isolation, quarantines, and lockdowns (*Mahikul et al., 2021*; *Triukose et al., 2021*; *Yorsaeng et al., 2022*). Previous studies have examined the association between preventive behaviors among older adults to protect themselves, focusing only on urban and high-epidemic settings (*Ounsaneha et al., 2023*; *Upake et al., 2022*; *Yodmai et al., 2021*). Previous studies focused only on urban and high-epidemic settings without specifying the regions or provinces covered. Therefore, our study, which assesses all regions of Thailand, provides a more comprehensive representation. Studies have revealed that good COVID-19 preventive behaviors are associated with factors including adequate income, convenient access to health services, knowledge, and strong family support (*Pechrapa et al., 2021*; *Yodmai et al., 2021*).

Preventive behaviors, such as wearing a mask, washing hands frequently, and maintaining social distance, are crucial in reducing COVID-19 transmission (*Doung-Ngern et al., 2020*; *Hamdan et al., 2023*; *Jang, Jang & Lee, 2020*; *Leech et al., 2022*). Major factors affecting COVID-19 prevention behaviors were predominantly age, sex (*Wachira et al., 2023*), perceived risk of exposure, knowledge and perception of personal preventability (*Limkunakul et al., 2022*), and health belief (*Aschwanden et al., 2021*; *Baghernezhad Hesary et al., 2021*; *Duan et al., 2022*; *Nanthamongkolchai et al., 2022*; *Rojpaisarnkit et al., 2022*; *Zewdie et al., 2022*). Older adults who live in the community and have underlying health conditions are particularly vulnerable to the virus, which increases the risk of severe illness or death (*Lekamwasam & Lekamwasam, 2020*; *Webb & Chen, 2022*). *Upake et al. (2022)* revealed that most participants in the study (approximately 70.8%) exhibited high levels of preventive behaviors toward COVID-19 in the Thung Phaya Thai and Nuan Chan districts in Bangkok, Thailand (*Upake et al., 2022*). The most influential factor in

determining these behaviors was an individual's belief in their ability to prevent COVID-19. Following this, COVID-19 response effectiveness, understanding of COVID-19, and sex were identified as other significant predictors. It is therefore essential to understand how well community-dwelling older adults in Thailand are adhering to COVID-19 preventive behaviors (*Upake et al., 2022*). To prevent the spread of COVID-19, it is crucial to promote adherence to preventive behaviors among community-dwelling older adults in Thailand (*Yodmai et al., 2021*).

Various theories and models have been suggested by sociologists, psychologists, and anthropologists to explain the factors affecting health behavior. This includes the health belief model (HBM), developed by Rosenstock and colleagues (*Rosenstock, 1974*; *Rosenstock, Strecher & Becker, 1988*). It serves as a general framework and theoretical guide for research in public health and comprises several components or constructs, such as perceived susceptibility, perceived severity, perceived benefits, perceived barriers, cues to action, and preventive health behaviors (*Rosenstock, 1974*; *Shahnazi et al., 2020*). The HBM provides a useful framework for designing effective interventions to encourage these behaviors (*Alagili & Bamashmous, 2021*; *Zewdie et al., 2022*). It considers individual perceptions of susceptibility, severity, benefits, barriers, and cues to action related to a health issue (*Zewdie et al., 2022*). By understanding these factors, interventions can be designed to address specific concerns and promote behavioral change (*Anagaw, Tiruneh & Fenta, 2023*). This approach can be particularly useful in Thailand, where cultural beliefs and social norms may affect the willingness of older adults to adopt preventive behaviors during the COVID-19 pandemic (*Upake et al., 2022*). Studies in Thailand have focused on village health volunteers (*Wongrith et al., 2024*), residents of Bangkok (*Janmaimool et al., 2024*), and the working-age Thai population (*Rojpaisarnkit et al., 2022*); however, no study has assessed the factors influencing preventive behaviors among community-dwelling older adults across all regions of Thailand. Studying all regions of Thailand allows for a comprehensive understanding of the preventive behaviors among older adults, leading to more effective public health strategies. It also helps develop nationwide strategies that address the specific needs of older populations across all communities.

This study aimed to examine preventive behaviors among community-dwelling older adults in all regions of Thailand during the COVID-19 pandemic. The study assessed the factors that may influence the adoption of these preventive behaviors, including age, sex, education level, and perceived risk of COVID-19. The findings of this study will provide valuable information for policymakers and healthcare providers on how to better support older adults in the community in adopting and maintaining preventive behaviors during future pandemics or new disease outbreaks.

## Significance for public health

This study has significant implications for public health in Thailand during the COVID-19 pandemic, especially for community-dwelling older adults. The lack of prior research on preventive behaviors among older adults across regions highlights a critical gap. By applying the HBM, our findings offer actionable insights into factors influencing COVID-19 preventive behaviors. The identified associations with knowledge, perceived susceptibility,
benefits, barriers, and cues to action underscore possible health education approaches for targeted interventions. Implementing educational campaigns based on these factors has the potential to enhance adherence to preventive measures by emphasizing benefits, overcoming barriers, and leveraging social media for health prevention. These strategic efforts could substantially improve the practices of older adults, fostering a more preventive community response to the ongoing pandemic in Thailand.

## MATERIALS AND METHODS

### Study design and participants
Between July and August 2021, a cross-sectional study was conducted with the participation of 910 community-dwelling older adults aged $\geq$60 years in Thailand. The inclusion criteria were as follows: age $\geq$60 years, having the ability to communicate, and providing voluntary consent to participate. The exclusion criteria included individuals with severe cognitive impairments such as dementia, mobility issues, and those unable to communicate.

### Sample size calculation
In a previous Thai study, 65.2% of residents had a high level of preventive measures (*Ounsaneha et al., 2023*). A prior sample size calculation was performed based on an anticipated effect size ($p = 0.652$), a desired precision level ($d = 0.034$), and a significance level of 0.05. To estimate the level of preventive measures, the formula for an infinite population proportion ($n = Z^2 p(1-p)/d^2$) was applied. This calculation ensured an adequate sample size to detect the level of preventive measures with a precision of 3.4% (d) and a 95% confidence level. Based on these parameters, the required sample size was estimated to be 852 participants. After accounting for a 20% non-response rate, the final sample size was adjusted to approximately 910 participants.

### Sampling
In this study, a rigorous recruitment process was employed to ensure the selection of a representative sample. A multistage stratified random sampling technique was used to select participants from each region. We used simple random sampling to select one province from each region, representing large cities with high population densities and large older adult populations (*Anantanasuwong, Pengpid & Peltzer, 2022*). The selected provinces were Chonburi (Eastern region), Chiang Mai (Northern region), Nakhon Sawan (Central region), Sakon Nakhon (Northeastern region), and Phatthalung (Southern region). Next, we applied simple random sampling to select one district from each province, followed by another round of simple random sampling to choose one subdistrict from each district. In the final stage, older adults were selected using simple random sampling without replacement with proportional allocation based on the distribution of older adults in each area. The total number of participants in the final sample was 910, distributed across regions as follows: Chonburi ($n = 204$), Chiang Mai ($n = 155$), Nakhon Sawan ($n = 241$), Sakon Nakhon ($n = 139$), and Phatthalung ($n = 171$).

## Data collection

Data were collected through a structured interview process with individual participants. The researcher coordinated with the director of the local hospital, also known as a subdistrict health-promoting hospital, which serves as a primary care unit located near households and provides care for older adults. The researcher also collaborated with health staff, including professional nurses, to explain the research objectives and procedures, obtain permission for data collection, and compile a list of older adults residing in the study area. Once the sample list was obtained, a professional nurse assisting with the research contacted selected participants to explain the study, obtain consent, and conduct a preliminary dementia screening Mini-Mental State Examination (MMSE) for dementia screening (*Tanglakmankhong et al., 2022*). If participants were eligible and agreed, a follow-up appointment was scheduled for data collection. The research team then visited participants at home, introduced themselves, reiterated the study's purpose, and conducted a brief 10–15 min interview after securing consent. To ensure COVID-19 safety, the team wore masks and face shields, provided masks for participants, and used hand sanitizer before and after each visit. Finally, the researcher reviewed the completed interview forms.

## Measures

The survey questionnaire was developed based on the literature review. The questionnaire used in this study comprised five distinct sections. These sections included inquiries about demographic characteristics, knowledge during the COVID-19 outbreak, questions related to the HBM constructs, and inquiries about preventive measures taken to mitigate the spread of COVID-19 using survey questionnaires from previous studies for items on susceptibility to infection, severity after infection, cues to action, and outbreak knowledge (*Bressington et al., 2020*), as well as for questions on perceived susceptibility, severity, cues to action, and knowledge (*Kwan et al., 2021*).

Demographic characteristics: This section of the questionnaire consisted of inquiries regarding several demographic factors, including age, sex, marital status, religion, education level, occupation, monthly income, underlying health conditions, and city of residence, as presented in Table 1. Knowledge during the COVID-19 outbreak: In this study, the participants were asked 12 questions aimed at assessing their ability to recognize clinical symptoms, prevention, and control of the disease. The questions were answered using "yes," "no," or "I don't know." A correct response was assigned a score of 1, whereas an incorrect or "I don't know" response was scored as 0. Maximum scores for knowledge were 12 points, as presented in Table 2. Preventive behaviors for COVID-19: The questionnaire included four inquiries regarding preventive measures taken to mitigate the spread of COVID-19. The participants were asked to rate their responses on a four-point Likert scale, ranging from "Never" to "Always," with scores ranging from 1 to 4. Total scores for preventive behaviors were 16 points, as presented in Table 3. HBM constructs: The questionnaire contained questions pertaining to various constructs of the HBM, which were divided into five sections. These sections included inquiries about perceived susceptibility, perceived severity, perceived benefits, perceived barriers, and cues to action and comprised four, four, five, five, and three questions, respectively. The participants were asked to rate

**Table 1 Demographic characteristics of participants ($N = 910$).**

| Variable | n | (%) |
|---|---|---|
| Sex | | |
| Female | 562 | 61.8 |
| Male | 348 | 38.2 |
| Age (years) | | |
| 60–69 | 678 | 74.5 |
| ≥70 | 232 | 25.5 |
| Marital status | | |
| Unmarried | 191 | 21.0 |
| Married | 480 | 52.7 |
| Divorced | 239 | 26.3 |
| Religion | | |
| Buddhist | 855 | 94.0 |
| Muslim | 42 | 4.6 |
| Christian | 13 | 1.4 |
| Education | | |
| Uneducated | 37 | 4.1 |
| Lower education (below bachelor's degree) | 821 | 90.2 |
| Bachelor's degree | 36 | 4.0 |
| Higher education (above bachelor's degree) | 16 | 1.7 |
| Occupation | | |
| Unemployed | 221 | 24.3 |
| Nongovernment employee | 642 | 70.5 |
| Government employee | 47 | 5.2 |
| Monthly income | | |
| <1,000 THB | 110 | 12.1 |
| 1,000 THB and over | 800 | 87.9 |
| Underlying disease | | |
| No | 197 | 21.6 |
| Yes | 713 | 78.4 |
| Region | | |
| Central | 241 | 26.5 |
| Northern | 155 | 17.0 |
| Northeastern | 139 | 15.3 |
| Eastern | 204 | 22.4 |
| Southern | 171 | 18.8 |

their responses on a four-point Likert scale, ranging from 1 = "Strongly disagree" to 4 = "Strongly agree" (Table 4).

## Validity and reliability

The original questionnaire was translated into Thai for use in our local setting by a research associate, while a second research associate was responsible for the back-translation into English. For content validation, three experts used a validation form to evaluate the

**Table 2 Responses to the questionnaire about knowledge during the COVID-19 outbreak among community-dwelling older adults in Thailand.**

| Statement | N (%) | |
| --- | --- | --- |
| | **Correct answer** | **Incorrect answer** |
| 1. Wearing a face mask and washing hands can prevent COVID-19. | 888 (97.6) | 22 (2.4) |
| 2. COVID-19 patients who cough or sneeze can easily transmit the disease. | 870 (95.6) | 40 (4.4) |
| 3. Immunity against COVID-19 can prevent this disease. | 759 (83.4) | 151 (16.6) |
| 4. Avoiding meeting people in crowded places can prevent COVID-19. | 859 (94.4) | 51 (5.6) |
| 5. COVID-19 patients might experience fever or chills, a dry cough, and shortness of breath. | 814 (89.5) | 96 (10.5) |
| 6. The symptoms of COVID-19 were less severe than the common cold. | 734 (80.7) | 176 (19.3) |
| 7. Self-prevention is better than cure. | 668 (73.4) | 242 (26.6) |
| 8. Older people are at higher risk of severe disease than other people. | 745 (81.9) | 165 (18.1) |
| 9. The transmission of SARS-CoV-2 happens via respiratory droplets, which are produced when individuals who are infected cough or sneeze. | 778 (85.5) | 132 (14.5) |
| 10. In the absence of a fever, individuals who are infected with SARS-CoV-2 cannot pass on the virus to others. | 518 (56.9) | 392 (43.1) |
| 11. Individuals who come into contact with a person who has contracted SARS-CoV-2 should promptly undergo a 14-day period of quarantine in a suitable location for general monitoring purposes. | 722 (79.3) | 188 (20.7) |
| 12. Effectively isolating and providing treatment to individuals who are infected with SARS-CoV-2 are measures that effectively minimize the transmission of the virus. | 761 (83.6) | 149 (16.4) |

**Table 3 Frequency distribution of conditions for observing COVID-19 preventive behaviors among community-dwelling older adults in Thailand.**

| Statement | N (%) | | | |
| --- | --- | --- | --- | --- |
| | **Always** | **Usually** | **Sometimes** | **Never** |
| 1. Wearing face masks while going to public areas. | 409 (44.9) | 459 (50.4) | 43 (4.7) | 0 (0.0) |
| 2. Washing hands with water using alcohol gel or soap before eating. | 508 (55.8) | 396 (43.5) | 6 (0.7) | 0 (0.0) |
| 3. Avoiding touching your eyes, nose, and mouth with hands. | 301 (33.1) | 584 (64.2) | 25 (2.7) | 0 (0.0) |
| 4. Meeting people in crowded places. | 339 (37.3) | 571 (62.7) | 0 (0.0) | 0 (0.0) |

relevance of each item to the intended construct. The content validity index was then calculated to measure the level of agreement among the experts regarding each item's relevance to the construct. The questionnaire's reliability was assessed among 30 older adults using the internal reliability method of Cronbach's α. The results indicated that
**Table 4 Responses to the questionnaire based on the health belief model constructs among community-dwelling older adults in Thailand.**

| Statement | N (%) | | | |
|---|---|---|---|---|
| | **Strongly agree** | **Partially agree** | **Partially disagree** | **Strongly disagree** |
| Perceived susceptibility | | | | |
| 1. I'm not concerned about COVID-19 and do my daily activities like before. | 0 (0.0) | 2 (0.2) | 177 (19.5) | 731 (80.3) |
| 2. I am more likely to get an infection. | 689 (76.7) | 211 (23.2) | 1 (0.1) | 0 (0.0) |
| 3. I consider myself to be at risk of COVID-19. | 627 (68.9) | 279 (30.7) | 4 (0.4) | 0 (0.0) |
| 4. My health may be at risk if I get the disease. | 609 (66.9) | 292 (32.1) | 9 (1.0) | 0 (0.0) |
| Perceived severity | | | | |
| 1. COVID-19 has a high death rate. It scares me. | 554 (60.9) | 326 (35.8) | 30 (3.3) | 0 (0.0) |
| 2. I am worried about the disease and cannot sleep well. | 520 (57.1) | 328 (36.1) | 62 (6.8) | 0 (0.0) |
| 3. I am worried about getting the disease because it can easily spread. | 681 (74.8) | 227 (25.0) | 2 (0.2) | 0 (0.0) |
| 4. I am worried about getting the disease because there is no effective vaccine or medicine. | 709 (77.9) | 195 (21.4) | 6 (0.7) | 0 (0.0) |
| Perceived benefits | | | | |
| 1. COVID-19 can be prevented easily with personal protective equipment, such as masks and disposable gloves. | 813 (89.3) | 97 (10.7) | 0 (0.0) | 0 (0.0) |
| 2. COVID-19 can be prevented easily by washing hands regularly with soap and water. | 654 (71.9) | 225 (28.0) | 1 (0.1) | 0 (0.0) |
| 3. COVID-19 can be prevented by avoiding meeting people in crowded places. | 482 (53.0) | 365 (40.1) | 63 (6.9) | 0 (0.0) |
| 4. COVID-19 can be prevented easily by keeping social distancing. | 443 (48.7) | 417 (45.8) | 50 (5.5) | 0 (0.0) |
| 5. COVID-19 can be prevented easily by taking a shower or bath. | 375 (41.2) | 486 (53.4) | 49 (5.4) | 0 (0.0) |
| Perceived barriers | | | | |
| 1. It is difficult to follow the instructions to prevent this disease. | 100 (11.0) | 239 (26.3) | 224 (24.6) | 347 (38.1) |
| 2. It is difficult to wash hands regularly with soap and water. | 118 (13.0) | 261 (28.7) | 206 (22.6) | 325 (35.7) |
| 3. To wear a surgical mask or cloth mask outside the household is difficult. | 101 (11.1) | 287 (31.5) | 162 (17.8) | 360 (39.6) |
| 4. It is easy to touch my mouth, nose, and eyes. | 175 (19.2) | 266 (29.3) | 182 (20.0) | 287 (31.5) |
| 5. Shopping at the market and having to prevent the disease are difficult. | 163 (17.9) | 345 (27.9) | 172 (18.9) | 230 (25.3) |
| Cues to action | | | | |
| 1. TV, Facebook, Line, and radio information about COVID-19 has been helpful. | 527 (57.9) | 346 (38.0) | 35 (3.9) | 2 (0.2) |
| 2. The recommendation to prevent the disease from healthcare workers has been helpful. | 545 (59.9) | 339 (37.3) | 25 (2.7) | 1 (0.1) |
| 3. The recommendation to prevent the disease from household members has been helpful. | 486 (53.4) | 382 (42.0) | 39 (4.3) | 3 (0.3) |

**Table 5  Frequency distribution of mean, standard deviation, standardized mean, and Cronbach's α of health belief model constructs among community-dwelling older adults in Thailand.**

| Statement | Number of questions | Range of scores for questions | Mean of total score | SD | Average score (ranging from 1 to 4) of each dimension | Minimum of total score | Maximum of total score | Cronbach's alpha |
|---|---|---|---|---|---|---|---|---|
| Perceived susceptibility | 4 | 1–4 | 14.94 | 1.38 | 3.73 | 4 | 16 | 0.705 |
| Perceived severity | 4 | 1–4 | 14.65 | 1.57 | 3.66 | 4 | 16 | 0.712 |
| Perceived benefits | 5 | 1–4 | 17.86 | 1.96 | 3.57 | 12 | 20 | 0.802 |
| Perceived barriers | 5 | 1–4 | 13.72 | 4.57 | 2.74 | 5 | 20 | 0.909 |
| Cues to action | 3 | 1–4 | 10.59 | 1.44 | 3.53 | 5 | 20 | 0.771 |

HBM included perceived susceptibility, perceived severity, perceived benefits, perceived barriers, and cues to action and had an internal consistency of Cronbach's $\alpha = 0.70$, 0.71, 0.80, 0.91, and 0.77, respectively (Table 5). An α value ranging from 0.6 to 0.7 indicates an acceptable level of reliability (*Taber, 2018*).

## Data analysis

The study population characteristics are expressed as frequencies and percentages for categorical data or as means and standard deviations for continuous data. Univariate analyses, including independent-samples t-tests or one-way ANOVA for parametric tests, were conducted to examine the associations between sociodemographic characteristics and practice scores. We used Pearson correlation coefficient analysis to examine the correlations between the HBM dimensions and preventive behaviors. A linear regression analysis was performed to investigate the factors, including sociodemographic characteristics and HBM dimensions, influencing preventive behaviors. Only variables that demonstrated a significant association with the COVID-19 preventive behaviors ($p$-value < 0.10) in the univariate analyses were included in the multiple linear regression model. The assumption of linearity was assessed by examining the residuals. Collinearity can reduce the accuracy of estimated coefficients. Therefore, linear relationships between continuous candidate predictors were evaluated using the variance inflation factor (VIF), with values $\geq 5$ considered problematic (*Deforth et al., 2022*). Additionally, Cook's distance was assessed, where a value less than 1—based on *Cook's (1977)* criterion—generally indicates that a data point in a regression analysis does not exert a strong influence on the model fit (*Cook, 1977*). The data were analyzed using SPSS software version 27.0 (IBM Corp., SPSS Statistics, Armonk, NY, USA) and statistical significance was set at $p$-value < 0.05.

## Ethical considerations

This study followed the ethical principles of the Declaration of Helsinki and was approved by the Burapha University Institutional Review Board (Number IRB1-050/2564). All participants were informed about the purpose of the study, and their participation was voluntary. Informed consent was obtained from all participants. Data confidentiality will be maintained throughout the study, and all personal information will be kept anonymous.

## RESULTS

### Demographic characteristics

The participants ($n = 910$) had a mean age with a standard deviation of $66.5 \pm 4.64$ years, with the majority being women (61.8%) and married (52.7%). Their education levels ranged from uneducated (4.1%) to higher education (above bachelor's degree) (1.8%). The majority of participants (78.4%) reported that they had underlying diseases. A total of 94.0% were Buddhist, 70.5% were nongovernment employees, 87.9% had a monthly income of 1,000 THB and over, and 26.5% lived in the Central region (Table 1).

### Knowledge

For knowledge regarding preventive behaviors (Table 2), 97.6% agreed that wearing a face mask and washing hands can prevent COVID-19, 95.6% agreed that COVID-19 patients who cough or sneeze can easily transmit the disease, and 94.4% agreed that avoiding meeting people in crowded places can prevent COVID-19. However, a proportion of participants (43.1%) misunderstood that people infected with SARS-CoV-2 cannot transmit the virus to others when a fever is not present, when in fact they can.

### Preventive behaviors

The participants' compliance with preventive behaviors related to COVID-19 was assessed, and the majority (55.8%) always washed their hands using alcohol gel or soap and cleaned with water before eating. Other frequently observed behaviors include wearing masks when going to public places (44.9%), and meeting people in crowded places (37.3%). The lowest levels of compliance (only 33.1%) were observed in relation to avoiding touching the eyes, nose, and mouth (Table 3).

### HBM constructs

According to the findings, the majority of participants showed high levels of perceived susceptibility, perceived severity, perceived benefits, and cues to action, but low levels of perceived barriers (Table 4). The findings revealed that, among the participants, 95.9% agreed that "TV, Facebook, Line, and radio information about COVID-19 has been helpful"; 100% agreed that "COVID-19 can be prevented easily with personal protective equipment, such as masks and disposable gloves"; and 99.9% agreed that "COVID-19 can be easily prevented by washing hands regularly with soap and water."

The mean total scores (with standard deviations) and the average scores (ranging from 1 to 4) for each dimension of the HBM constructs are presented in Table 5. The construct with the highest average score was "perceived susceptibility" (3.73), followed by "perceived severity" (3.66), "Perceived benefits" (3.57), and "cues to action" (3.53). The lowest mean scores were observed for "perceived barriers" (2.74).

Table 6 indicates that COVID-19 preventive behaviors were positively and significantly correlated with several HBM constructs, including perceived benefits ($r = 0.258$), internal cues to action ($r = 0.209$), perceived susceptibility ($r = 0.100$), and knowledge ($r = 0.177$). In contrast, perceived barriers had a significant negative correlation with COVID-19 preventive behaviors ($r = -0.157$; $p < 0.05$).

**Table 6 Correlations between preventive behaviors and health belief model construct from COVID-19 among community-dwelling older adults in Thailand.**

| Statement | Perceived susceptibility | Perceived severity | Perceived benefits | Perceived barriers | Cues to action | Knowledge | Preventive behaviors |
|---|---|---|---|---|---|---|---|
| Perceived susceptibility | 1 | | | | | | |
| Perceived severity | 0.172** | 1 | | | | | |
| Perceived benefits | 0.061 | 0.077* | 1 | | | | |
| Perceived barriers | −0.086** | −0.049 | −0.098** | 1 | | | |
| Cues to action | 0.053 | 0.031 | 0.375** | 0.007 | 1 | | |
| Knowledge | 0.006 | 0.020 | 0.305** | 0.021 | 0.254** | 1 | |
| Preventive behaviors | 0.100** | 0.055 | 0.258** | −0.157** | 0.209** | 0.177** | 1 |

Notes.
*$p < 0.05$ (two-tailed).
**$p < 0.01$ (two-tailed).

**Table 7 Association between demographic characteristics and practice score and during COVID-19 among community-dwelling older adults in Thailand.**

| Variable | Category | Practice score (Mean ± SD) | p value |
|---|---|---|---|
| Sex | Female | 13.64 ± 1.33 | 0.740 |
| | Male | 13.61 ± 1.34 | |
| Age (years) | 60–69 | 13.67 ± 1.33 | 0.165 |
| | ≥70 | 13.53 ± 1.36 | |
| Marital status | Unmarried | 13.68 ± 1.34 | 0.586 |
| | Married | 13.59 ± 1.35 | |
| | Divorced | 13.68 ± 1.30 | |
| Religion | Buddhist | 13.61 ± 1.34 | 0.083 |
| | Muslim | 14.07 ± 1.11 | |
| | Christian | 13.46 ± 1.45 | |
| Education | Uneducated | 13.54 ± 1.43 | 0.124 |
| | Lower education (below bachelor's degree) | 13.66 ± 1.33 | |
| | Bachelor's degree | 13.19 ± 1.21 | |
| | Higher education (above bachelor's degree) | 13.25 ± 1.39 | |
| Occupation | Unemployed | 13.51 ± 1.35 | 0.307 |
| | Nongovernment employee | 13.67 ± 1.33 | |
| | Government employee | 13.64 ± 1.33 | |
| Monthly income | <1,000 THB | 13.56 ± 1.42 | 0.574 |
| | 1,000 THB and over | 13.64 ± 1.32 | |
| Underlying disease | No | 13.62 ± 1.28 | 0.892 |
| | Yes | 13.63 ± 1.35 | |

Notes.
*$p < 0.05$ using the Student $t$-test or one-way ANOVA.

No significant difference was observed between demographic characteristics and preventive behaviors, as presented in Table 7.

The analysis indicated that perceived susceptibility, perceived benefits, perceived barriers, cues to action, and knowledge were significant factors affecting preventive behaviors against

**Table 8  Effects of constructs of the health belief model on COVID-19 preventive behaviors among community-dwelling older adults in Thailand.**

| Statement | Univariate analysis | | | | Multiple analysis | | | |
|---|---|---|---|---|---|---|---|---|
| | Estimation (β) | Standard error | Confidence interval | p value | Estimation (β) | Standard error | Confidence interval | p value |
| Gender | −0.030 | 0.091 | −0.209, 0.148 | 0.740 | | | | |
| Age | −0.005 | 0.010 | −0.024, 0.014 | 0.591 | | | | |
| Marital status | 0.004 | 0.065 | −0.123, 0.131 | 0.950 | | | | |
| Religion | 0.159 | 0.141 | −0.119, 0.436 | 0.262 | | | | |
| Education | −0.181 | 0.114 | −0.405, 0.044 | 0.115 | | | | |
| Occupation | 0.114 | 0.087 | −0.057, 0.285 | 0.191 | | | | |
| Monthly income | 0.076 | 0.136 | −0.190, 0.434 | 0.574 | | | | |
| Underlying disease | 0.015 | 0.107 | −0.196, 0.226 | 0.892 | | | | |
| Perceived susceptibility | 0.097 | 0.032 | 0.034, 0.159 | 0.002 | 0.066 | 0.031 | 0.006, 0.127 | 0.032 |
| Perceived severity | 0.047 | 0.028 | −0.008, 0.102 | 0.097 | 0.016 | 0.027 | −0.038, 0.069 | 0.565 |
| Perceived benefits | 0.175 | 0.022 | 0.132, 0.218 | <0.001 | 0.111 | 0.024 | 0.064, 0.158 | <0.001 |
| Perceived barriers | −0.046 | 0.010 | −0.065, −0.027 | <0.001 | −0.040 | 0.009 | −0.058, −0.023 | <0.001 |
| Cues to action | 0.194 | 0.030 | 0.135, 0.253 | <0.001 | 0.110 | 0.032 | 0.048, 0.173 | <0.001 |
| Knowledge | 0.163 | 0.030 | 0.104, 0.222 | <0.001 | 0.091 | 0.031 | 0.031, 0.152 | 0.003 |

**Notes.**
*$p < 0.05$.

COVID-19, according to the univariate and multiple regression analysis (Table 8). No significant difference was observed between demographic characteristics and preventive behaviors in the univariate analysis. Therefore, demographic characteristics were excluded from the regression analysis. The results of the multiple regression analysis revealed that COVID-19 preventive behaviors were significantly related perceived susceptibility ($\beta = 0.066$, $p$-value = 0.032), perceived benefits ($\beta = 0.111$, $p$-value < 0.001), perceived barriers ($\beta = -0.040$, $p$-value < 0.001), cues to action ($\beta = 0.110$, $p$-value < 0.001), and knowledge ($\beta = 0.091$, $p$-value = 0.003), except for perceived severity ($\beta = 0.016$, $p$-value < 0.565). Moreover, perceived susceptibility and perceived benefits were positively correlated with COVID-19 preventive behaviors, meaning that higher scores on these constructs were associated with higher performance. However, perceived barriers were negatively correlated and resulted in lower performance.

## DISCUSSION

The results revealed significant relationships between COVID-19 preventive behaviors and factors such as perceived susceptibility, perceived benefits, perceived barriers, cues to action, and knowledge (all $p < 0.01$). While demographic characteristics were not significantly associated with preventive behaviors during the COVID-19 pandemic. Specifically, higher levels of perceived susceptibility and benefits were associated with improved preventive behaviors, whereas perceived barriers correlated with lower performance in these behaviors. Enhancing individuals' sense of susceptibility and the benefits of preventive actions and minimizing perceived barriers can effectively promote COVID-19 preventive behaviors.

In this study, we investigated the factors influencing preventive behaviors among community-dwelling older adults in Thailand during the COVID-19 pandemic using the HBM as a framework. The key findings indicate that several dimensions of the HBM—perceived susceptibility, perceived benefits, and cues to action—significantly shaped preventive behaviors in this population. These results underscore the importance of tailored public health interventions that address these factors to enhance preventive practices among older adults, ultimately helping mitigate the impact of COVID-19 on this vulnerable group. However, demographic characteristics were not significantly associated with preventive behaviors, which aligns with some previous studies that found no significant associations between demographic factors such as age and income and preventive behaviors (*Li et al., 2022*). This suggests that individuals, regardless of their demographic characteristics, are equally concerned about engaging in preventive behaviors. In Thailand, as in other countries, the government and public health authorities applied various measures to prevent the spread of the virus, including social distancing, wearing masks, and frequent hand washing (*Mahikul et al., 2021*; *Triukose et al., 2021*; *Yorsaeng et al., 2022*). However, adherence to these preventive measures may vary depending on individual beliefs. Our findings indicated a satisfactory adherence rate to these measures, with good compliance, such as consistently washing hands before eating. Other frequently observed behaviors include wearing masks while going to public areas, and avoiding meeting people in crowded places. Similarly, a Hong Kong study reported that over 77% of the participants had good health performance regarding COVID-19 (*Kwok et al., 2020*) and in Iran, most participants (96.8%) refrained from going to crowded places to prevent the spread of the disease (*Shahnazi et al., 2020*).

The HBM is a widely used theoretical framework in health psychology for understanding the factors influencing health-related behaviors, including preventive behaviors during the COVID-19 pandemic (*Karimy et al., 2021*; *Kim & Kim, 2020*). Using the HBM, researchers in Thailand have investigated the factors influencing preventive behaviors during the COVID-19 pandemic among community-dwelling older adults. The study suggests that a community-dwelling older adult's decision to adopt health behaviors is influenced by their perceived susceptibility to a health problem, the perceived benefits and cues to action of taking preventive action, the perceived barriers to taking preventive action, and knowledge regarding preventive behaviors. Overall, our findings were consistent with those of previous research (*Chen et al., 2020*; *Kim & Kim, 2020*; *Smail et al., 2021*; *Upake et al., 2022*). The study's findings suggested that the HBM could explain 13% of the variation in COVID-19 preventive behaviors. These findings are consistent with some previous studies (*Karimy et al., 2021*; *Mirzaei et al., 2021*). An and colleagues reported that perceived susceptibility was the strongest predictor of behavioral change during the early stages of the COVID-19 pandemic in Italy and South Korea (*An, Schulz & Kang, 2023*). This could be due to a more protective intention to take preventive actions a few months after the COVID-19 outbreak.

We also found a significant relationship between perceived susceptibility and the adoption of COVID-19 preventive behaviors. Individuals need to regard themselves as vulnerable to the disease and recognize its severity as a substantial threat (*DeDonno et al., 2022*). In an early-stage study of COVID-19 in Hong Kong, a substantial majority (89%)
of individuals exhibited a heightened perception of their susceptibility to COVID-19, indicating that they considered themselves at risk for the disease (*Kwok et al., 2020*), compared with 76.7% of our participants. Overall, the perceived severity of COVID-19 was higher than that of SARS. Some studies have shown that individuals who perceive less susceptibility to the disease may consider it a severe disease (*Malhotra et al., 2016*; *Shahnazi et al., 2020*).

The perception of benefits was found to be another essential factor in predicting preventive behaviors against the disease. In other words, individuals were more likely to engage in preventive measures when they perceived the benefits of these actions to be high. For instance, understanding the positive impact of regular handwashing and using personal protective equipment, such as masks and disposable gloves, can amplify the perceived benefits. Our findings agree with those of previous research (*Fathian-Dastgerdi et al., 2021*; *Shah et al., 2021*).

Adherence to COVID-19 preventive behaviors increased when perceived barriers decreased. Overcoming perceived barriers is important because barriers can deter individuals from engaging in preventive behavior despite their desire to do so. In the study, the participants had fewer perceived barriers to individual preventive behaviors, such as touching hands, mouth, nose, and eyes. Numerous studies have demonstrated that respiratory infections can be transmitted *via* contaminated hands and objects (*Wanjari et al., 2023*), indicating that individuals may be at a greater risk of contracting COVID-19 if they touch their nose, mouth, or eyes after coming into contact with contaminated items (*Arceo et al., 2021*; *Scott, 2013*). Addressing these barriers may help improve adherence to preventive measures among older adults in Thailand.

Knowledge regarding preventive behaviors and education level were found to be an essential factor in predicting preventive behaviors against COVID-19 (*Maracy, Rahimi & Shahraki, 2020*). Participants who scored higher on knowledge related to COVID-19 also showed higher adherence to preventive behaviors against the disease (*Hosen et al., 2021*; *Pothisa et al., 2022*). However, a proportion of participants (43.1%) misunderstood that people infected with SARS-CoV-2 cannot transmit the virus to others when a fever is not present, when in fact they can. This misconception likely persists due to misinformation or lack of awareness about the asymptomatic transmission of COVID-19 (*Okereke et al., 2020*), which has been a central challenge in public health communication throughout the pandemic. Many individuals may associate fever with the presence of the virus, as it is a common symptom, leading to the false belief that those without fever are not contagious. Overall, the study highlights the importance of understanding the factors influencing preventive behaviors during the COVID-19 pandemic among community-dwelling older adults in Thailand. Using the HBM can provide a useful framework for developing interventions aimed at promoting and sustaining these behaviors. The HBM is currently used as a theoretical framework in health psychology that can be useful for understanding the factors influencing preventive behaviors during the COVID-19 pandemic. By focusing on community-dwelling older adults, the study may provide insights into the specific challenges and barriers faced by this population in adopting preventive behaviors. The study's findings may help inform the development of interventions aimed at promoting

and sustaining preventive behaviors among older adults in Thailand. Overall, studies that investigate the factors influencing preventive behaviors during the COVID-19 pandemic using the HBM can provide valuable insights into the challenges and barriers faced by different populations in adopting these behaviors. However, it is important to acknowledge and address the limitations of these studies to ensure that the findings are valid and applicable to different populations and contexts.

## Limitations

This study has some limitations. First, our findings may not be generalizable to other populations or contexts beyond community-dwelling older adults in Thailand. Second, the cross-sectional design of the study precluded the establishment of causal relationships. Third, we did not explore the influence of social or cultural factors that may affect preventive behaviors during the COVID-19 pandemic among community-dwelling older adults in Thailand. Fourth, the sample demographics reveal significant imbalances, with most participants being women, having lower education, higher income, and underlying diseases. As a result, the findings may not accurately represent the broader older adult population in Thailand. Fifth, the sample could not represent individuals at the national level because it was drawn from some provinces.

## CONCLUSIONS

Our findings indicate that having knowledge regarding preventive behaviors, perceived susceptibility, perceived benefits, perceived barriers, and cues to action were the strongest predictors of preventive behaviors against COVID-19. By highlighting the benefits of preventive behaviors, perceived susceptibility and perceived benefits can be increased, leading to the overcoming of barriers to such behaviors. Additionally, reducing perceived barriers and focusing more on promoting preventive behaviors are suggested. To improve the practices of community-dwelling older adults during the COVID-19 pandemic in Thailand, health information campaigns should focus on highlighting the advantages of preventive behaviors, tackling obstacles, offering suggestions to overcome them, displaying cues to action as reminders on preventive health social media, and increasing awareness about disease prevention and control. These could help reduce the chances of or mitigate a COVID-19 outbreak in the future. The study's findings can inform public health policies by integrating behavioral predictors into health campaigns, addressing perceived barriers, and enhancing emergency preparedness. In clinical practice, healthcare providers should incorporate behavioral insights into education and personalized interventions among community-dwelling older adults to improve adherence to preventive measures in the next emerging infectious disease outbreak.

## ACKNOWLEDGEMENTS

We are grateful to the participants and the staff for their assistance in conducting this study. We also thank Sudeep Agarwal from Scribendi for editing a draft of this manuscript. The author acknowledges that certain parts of this article were generated by ChatGPT

(powered by OpenAI's language model GPT-4.0; https://chatgpt.com/). Editing was done by the author.

### Funding
This work was funded by Burapha University, Thailand. This research project is supported by Chulabhorn Royal Academy. The funders had no role in study design, data collection and analysis, decision to publish, or preparation of the manuscript.

### Grant Disclosures
The following grant information was disclosed by the authors:
Burapha University, Thailand.
Chulabhorn Royal Academy.

### Competing Interests
The authors declare there are no competing interests.

### Author Contributions
- Kanchana Piboon conceived and designed the experiments, performed the experiments, analyzed the data, authored or reviewed drafts of the article, and approved the final draft.
- Jarinthip Chomchaipon performed the experiments, authored or reviewed drafts of the article, and approved the final draft.
- Dhammawat Ouppawongsapat performed the experiments, authored or reviewed drafts of the article, and approved the final draft.
- Wanlop Jaidee performed the experiments, authored or reviewed drafts of the article, and approved the final draft.
- Patchana Hengboriboonpong Jaidee performed the experiments, authored or reviewed drafts of the article, and approved the final draft.
- Paiboon Pongsaengpan performed the experiments, authored or reviewed drafts of the article, and approved the final draft.
- Wiriya Mahikul conceived and designed the experiments, performed the experiments, analyzed the data, prepared figures and/or tables, authored or reviewed drafts of the article, and approved the final draft.

### Human Ethics
The following information was supplied relating to ethical approvals (*i.e.*, approving body and any reference numbers):

This study followed the ethical principles of the Declaration of Helsinki and was approved by the Burapha University Institutional Review Board (Number IRB1-050/2564). All participants were informed about the purpose of the study, and their participation was voluntary. Informed consent was obtained from all participants. Data confidentiality will be maintained throughout the study, and all personal information will be kept anonymous.

## Data Availability

The raw data are available as Supplemental Files.

## Supplemental Information

Supplemental information for this article can be found online at http://dx.doi.org/10.7717/peerj.19412#supplemental-information.

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
