# Peer review of "Preventive behaviors of COVID-19 during the COVID-19 pandemic among community-dwelling older adults in Thailand"

_PeerJ, doi:10.7717/peerj.19412_

## Round 0.1 · original submission · Major Revisions

Your manuscript requires major revisions before further consideration. Please address each reviewer's comments point by point and submit a revised version with changes highlighted. We look forward to your resubmission.

Reviewer 1 ·

Basic reporting

The manuscript needs some corrections.

Experimental design

The sampling needs more explanations.

Validity of the findings

The tool was not explained properly.

Additional comments

The text needs editing.

Annotated reviews are not available for download in order to protect the identity of reviewers who chose to remain anonymous.

Reviewer 2 ·

Basic reporting

Overall the manuscript is written clearly and is a an easy read.

There is a good introduction and the analhsis make use of what appears to be solid global data.

The Health Belief Model (HBM) is a good conceptual foundation.

References are plentiful and current for the most part.

The data structure seems to be in good shape.

The STROBE checklist seems to indicate that the manuscript covers all the requirements.

The survey seems to have been well-structured, although linking it more directly to the HBM would be helpful.

Experimental design

The research methodology leaves something to be desired. Relevant comments follow.

In lines 201-210: The statistical discussion is jumbled, with several apparently contradictory methods mentioned that are appropriate to categorical and continuous data.

Using pairwise correlations and regressions to select predictors for the full model increases the risk of Type I error and biases the inclusion process relative to what could have been obtained through some version of stepwise processes by circumventing leverage plots, variance inflation factors, and other indicators that would be helpful in model building. This lack of procedural rigor leads, among other consequences, to the exclusion of all demographic indicators, which almost certainly biases the reported results.

The sex imbalance in the data raises the question of how different the results might be by using post-stratification to gain better demographic representation and then to re-estimate the prediction model and see if interpretations of the results are perceptibly different. Doing so would provide a useful test of the robustness of the reported findings.

In lines 265-266 the manuscript states: “To make the mean importance of each dimension comparable among the participants, the measured mean was obtained by dividing the mean score by the number of questions.” What was done when data were missing? This is a regular issue in SPSS, with the need to reduce the denominator due to item nonresponse to generate comparable scores.

Replication of the study would be difficult given the lack of clarity about research procedures.

Validity of the findings

Lines 401-411 provide some useful general suggested implications. Further elaboration on implications for health policy making and implementation, and for clinical practice, would strengthen the manuscript.

In Table 5 most of the reported values of Cronbach’s alpha reliability estimates are quite low, indicating the absence of cohesion among the bundled items. This lack of reliability further implies the probable lack of validity of the resulting summary metrics. That circumstances may help to explain the low correlations reported in Table 6.

Reviewer 3 ·

Basic reporting

no comment

Experimental design

no comment

Validity of the findings

no comment

Additional comments

A review of the manuscript entitled “preventive behaviors of covid-19 during the 1 covid-19 pandemic among community-dwelling older adults in thailand”
- (page 5, abstract’s methods) it is unclear how participants were selected (random sampling, convenience sampling, etc.), which is important for understanding the study's representativeness.
- (page 6, lines 61-63) the authors stated in their introduction that previous studies focused only on urban and high-epidemic settings but did not specify which regions or provinces those studies covered. Please provide a more explicit comparison between urban-focused studies and your research on all regions of Thailand to strengthen your argument.
- (page 6, lines 66-67) this sentence “Preventive behaviors, such as wearing a mask, washing hands frequently, and maintaining social distance, are crucial in reducing COVID-19 transmission.” also shares the same idea with a similar study by Hamdan et al. https://narrax.org/main/article/view/71/49 Kindly include this reference to make the sentence stronger.
- (page 6, line 76) the claim that "70.8% exhibited high levels of preventive behaviors" (from Upake et al., 2022) might overgeneralize findings from a specific study. If this was based on a regional or limited sample, clarify its scope to avoid misleading conclusions.
- (page 7, line 99) while the authors state that no study has assessed preventive behaviors among older adults across all Thai regions, it would help to explicitly highlight how this gap undermines public health strategies.
- (page 7, lines 125-127) the authors mentioned inclusion criteria (age ≥ 60 years, ability to communicate, voluntary consent), but there is no mention of exclusion criteria. For example, were individuals with severe cognitive impairments, mobility issues, or those already diagnosed with COVID-19 excluded?
- (page 7, line 132) the effect size (Cohen’s f = 0.652) seems incorrect for a “moderate effect size.” According to Cohen’s guidelines, an f value of 0.652 is considered large, not moderate. Recheck this value and ensure it aligns with your cited study.
- (page 8, line 156) the authors referenced a preliminary dementia screening but did not specify the tool or criteria used to assess cognitive function. Including this information is crucial to ensure the study's replicability and credibility.
- (page 9, line 207) while the threshold for significance is appropriately set at p < 0.05, it’s unclear if you adjusted this value for multiple comparisons (e.g., Bonferroni correction) during the regression analysis. Without this, there’s a risk of inflated Type I error.
- (page 9, line 222) the authors reported that 21.0% of participants were married. This seems unusually low given the demographic. If correct, specify whether the remaining participants were widowed, divorced, or single.
- (page 11, discussion) although the authors noted demographic characteristics were not significantly related to preventive behaviors, they didn’t discuss potential reasons for this finding or compare it with similar studies. Please include this to give valuable context.
- (page 12, discussion) it is noted that 43.1% of participants misunderstood that people without fever could not transmit the virus. This is a critical finding but lacks discussion on why this misconception persists and how it can be addressed (e.g., through targeted education). Please explain.
- (page 13, lines 368-371) this sentence “Numerous studies have demonstrated that respiratory infections can be transmitted via contaminated hands and objects, …” will be more significant if it’s also supported by another relevant study. For example, https://pubmed.ncbi.nlm.nih.gov/38454970/

Reviewer 4 ·

Basic reporting

no comment

Experimental design

1. The Cronbach's alpha values for perceived susceptibility (α = 0.574) and perceived severity (α = 0.523), are below the standard threshold of 0.70 required for acceptable internal consistency in scientific research. These low values raise significant questions about the measurement reliability.

2. The sampling approach presents additional methodological challenges. While the authors employed multistage stratified random sampling, the resulting sample demographics show notable imbalances. The data reveals disproportionate representation in several key categories: lower education (90.2%), higher income (87.9% above 1,000 THB), presence of underlying diseases (78.4%), and female participants (61.8%). These sampling skews limit the study's ability to represent Thailand's broader older adult population accurately.

3. The demographic categorization system requires significant improvement. The religious classification, which only includes Buddhist (94.0%), Muslim (4.6%), and Christian (1.4%). I am wondering why there is no participants with no religion, or with other religion. The education variable's broad grouping of "Lower education" (90.2%) combines distinctly different educational levels, potentially masking important variations in health literacy. Similarly, the binary income classification (<1,000 THB vs. ≥1,000 THB) lacks the necessary detail for meaningful socioeconomic analysis.

Validity of the findings

The study's validity is compromised by several limitations in the data presentation and analysis. The oversimplified demographic categorizations significantly restrict the examination of socioeconomic and cultural factors affecting preventive behaviors. The binary income classification makes it impossible to analyze how economic gradients influence adherence to preventive measures, while the broad education category potentially obscures important relationships between educational attainment and health behaviors.

To enhance the study's scientific rigor, I recommend using more nuanced demographic categories, particularly for education and income variables, and conducting sensitivity analyses to assess how the current categorical simplifications might affect the reported findings.

---

## Round 0.2 · accepted · Accept

Authors have addressed all of the reviewers' comments and manuscript is ready for publication.

Reviewer 2 ·

Basic reporting

The figures and tables seem fine.

The Excel file appears to be in good shape and should be very helpful for replication/extension studies.

Language generally is in good shape, with some minor editing needed to bring the documentup to standards for a major refereed journal.

The changes that were made following reviewer recommendations are reasonably thorough and resonsive to what was suggested.

Experimental design

The statistical strength of the study has been much improved.

Validity of the findings

The findings now seem more valid, given the reported new metrics (such as variance inflation factors). Also, realiability has been improved with items deleted where necessary.

Additional comments

The revisions seems fairly extensive and certainly have strengthened the document.

Reviewer 3 ·

Basic reporting

no comment

Experimental design

no comment

Validity of the findings

no comment

Additional comments

no comment

Reviewer 4 ·

Basic reporting

No additional comments.

Experimental design

Thank you for addressing the reliability concerns by recalculating Cronbach's alpha values after removing problematic items. And thank you for your comments regarding the sampling imbalances. For demographic categorization, I understand your explanations regarding income classification for elderly populations.

Validity of the findings

No additional comments.